# Ultra-Low Pt Loading in PtCo Catalysts for the Hydrogen Oxidation Reaction: What Role Do Co Nanoparticles Play?

**DOI:** 10.3390/nano11113156

**Published:** 2021-11-22

**Authors:** Felipe de Jesús Anaya-Castro, Mara Beltrán-Gastélum, Omar Morales Soto, Sergio Pérez-Sicairos, Shui Wai Lin, Balter Trujillo-Navarrete, Francisco Paraguay-Delgado, Luis Javier Salazar-Gastélum, Tatiana Romero-Castañón, Edgar Reynoso-Soto, Rosa María Félix-Navarro, Moisés Israel Salazar-Gastélum

**Affiliations:** 1Tecnológico Nacional de México, Instituto Tecnológico de Tijuana, Centro de Graduados e Investigación en Química, Tijuana 22510, Mexico; felipe0757@hotmail.com (F.d.J.A.-C.); mara.beltran@tectijuana.edu.mx (M.B.-G.); sperez@tectijuana.mx (S.P.-S.); sl388@aol.com (S.W.L.); balter.trujillo@tectijuana.edu.mx (B.T.-N.); luis.salazarg@tectijuana.edu.mx (L.J.S.-G.); edgar.reynoso@tectijuana.edu.mx (E.R.-S.); 2Tecnológico Nacional de México, Instituto Tecnológico de Tijuana, Posgrado en Ciencias de la Ingeniería, Tijuana 22510, Mexico; omar.soto18@tectijuana.edu.mx; 3Centro de Investigación en Materiales Avanzados S.C., Laboratorio Nacional de Nanotecnología, Chihuahua 31136, Mexico; francisco.paraguay@cimav.edu.mx; 4Instituto Nacional de Electricidad y Energías Limpias, Cuernavaca 62490, Mexico; tromero@ineel.mx

**Keywords:** hydrogen oxidation reaction, anodic catalyst, nanomaterial, anionic exchange membrane fuel cell, hydrogen pump, energy storage

## Abstract

The effect of the nature of the catalyst on the performance and mechanism of the hydrogen oxidation reaction (HOR) is discussed for the first time in this work. HOR is an anodic reaction that takes place in anionic exchange membrane fuel cells (AEMFCs) and hydrogen pumps (HPs). Among the investigated catalysts, Pt exhibited the best performance in the HOR. However, the cost and the availability limit the usage. Co is incorporated as a co-catalyst due to its oxophylic nature. Five different PtCo catalysts with different Pt loading values were synthesized in order to decrease Pt loading. The catalytic activities and the reaction mechanism were studied via electrochemical techniques, and it was found that both features are a function of Pt loading; low-Pt-loading catalysts (Pt loading < 2.7%) led to a high half-wave potential in the hydrogen oxidation reaction, which is related to higher activation energy and an intermediate Tafel slope value, related to a mixed HOR mechanism. However, catalysts with moderate Pt loading (Pt loading > 3.1%) exhibited lower E_1/2_ than the other catalysts and exhibited a mechanism similar to that of commercial Pt catalysts. Our results demonstrate that Co plays an active role in the HOR, facilitating H_ads_ desorption, which is the rate-determining step (RDS) in the mechanism of the HOR.

## 1. Introduction

The main strategy for the energy policies around the world consists in the incorporation of clean electrical energy from renewable resources in the electrical energy network. However, the generation rate from renewable resources is relatively unpredictable, depending upon several factors, such as the season, the weather, orography, and the availability of the resource (tidal, wind, solar, chemical, etc.) [1,2]. Supply of renewable energy to an electrical grid is a difficult task, since the generation/conversion rate does not usually match the requirements of the grid. In this regard, storage of the surplus electricity is important for the management of electrical energy supply. Electrical energy cannot be stored for long periods of time, since the time constant of the devices (capacitors and supercapacitors) are in the range from 10^−2^ to 10^2^ s.

Alternatively, electrical energy can be transformed into mechanical and chemical energies and stored for later use. From a mechanical point of view, compressed air and hydro storage systems are the main options commercially available [3,4]. However, the storage of electrical energy by either the chemical or the electrochemical route involves several devices, such as batteries, electrolyzers, and fuel cells [5,6,7].

The worldwide economy can be sustained by hydrogen, since hydrogen production is related to different technologies, such as (i) fossil/biomass combustion, (ii) chemical processes, (iii) thermolysis, etc. Moreover, water acts as a large reservoir of hydrogen, and electrolysis is a suitable method for hydrogen gas production that has gained the attention of the industrial and scientific community in recent years [8,9]. Alternatively, hydrogen pumps (HPs) are devices that can separate and concentrate hydrogen from a gas mixture coming from reformatting operations by redox processes. Hydrogen is considered an inexhaustible reservoir of clean energy with minimal impact on the environment.

Either in fuel cells or in HPs, hydrogen is selectively oxidized at the anode. Only heat and water are the main byproducts of this energy conversion process in the fuel cell, while purified hydrogen gas is obtained at the HP. A key factor when the efficiency of the process in fuel cells and HPs is compared against that in other devices, such as an internal combustion engine, is that the fuel cell and the HP are not limited by the Carnot cycle [10].

There are several technologies related to the fuel cell, such as the solid oxide fuel cell (SOFC), the enzymatic fuel cell (EFC), the proton exchange membrane fuel cell (PEMFC), and the anionic exchange membrane fuel cell (AEMFC) [11,12,13,14]. PEMFC and AEMFC are attractive options because the operation temperature of these fuel cells is below 100 °C and because of their scalability. Moreover, both technologies have flexibility in the use of fuels, such as hydrogen, alcohols, ammonia (only for the AEMFC), and low-hydrocarbon compounds. However, hydrogen has minimal impact on the environment, unlike other fuels.

The reactions that take place in these fuel cells are the oxygen reduction reaction (ORR) and the hydrogen oxidation reaction (HOR), in the cathode and anode compartments, respectively. The main difference between the AEMFC and the PEMFC is in the transfer of ions in the process. Pt-based materials are the most-reported catalysts for both reactions. An advantage of the AEMFC over the PEMFC is that the cathodic catalyst could be non-expensive metals instead of Pt under alkaline conditions. Although Pt shows the best performance as a catalyst in the anodic reaction, the cost and availability of this metal are the main drawbacks in the commercialization of the fuel cell technology [15,16,17]. Additionally, the kinetic rate constant is slower by 2 orders of magnitude in alkaline conditions compared to acidic conditions [18,19]. Many strategies have been reported in the area of decreasing the Pt loading on the anodic and the cathodic catalysts of the fuel cell to increase the catalytic activity, such as (i) the use of Pt nanoparticles dispersed in carbon supports, such as graphene, amorphous carbon, and carbon nanotubes, and/or (ii) the use of Pt alloys with non-expensive metals.

Regarding the operation of the HP, the aforementioned HOR takes place at the anode, while the hydrogen evolution reaction (HER) occurs at the cathode, where the theoretical standard cell potential is 0.0 V vs. SHE, dominating the resistive behavior of the membrane instead of the activation and ohmic overpotential during the HP operation [20]. Nevertheless, there is essentially an affordable low-activation-energy anodic catalyst.

Another key aspect about an anodic catalyst for AEMFC and HP devices is the understanding of the mechanism of the HOR, which is explained mainly by Koutecky–Levich analysis. Since the HOR is a limiting-diffusion process, the inverse of current density varies linearly to the inverse of the square root of the rotation rate of the electrode, under hydrodynamic conditions, according to Equation (1):(1)$$\frac{1}{j}=\frac{1}{j_{k}}+\frac{1}{j_{d}}=\frac{1}{j_{k}}+\frac{1}{BC_{0}\omega^{1/2}}$$
where B is the Levich coefficient, which implies the number of electrons transferred during the HOR. Zhuang and coworkers reported the Levich coefficient for the Pt polycrystalline to be 4.87 cm^2^ mA^−1^ s^−1/2^ associated with two electrons transferred [18,21].

The rate-determining step (RDS) is associated with the Tafel slope value. Catalysts with a Tafel slope value of 30 mV dec^−1^ are related to the dissociative hydrogen adsorption as the RDS (Tafel step; Equation (2)), while catalysts that exhibit a Tafel slope value of 120 mV dec^−1^ are associated with electronic charge transfer (via either the Heyrovsky step or the Volmer step; Equations (3) and (4), respectively) [10,22,23].
(2)$$H_{2}+2M⇄2M-H_{\mathrm{ads}}$$
(3)$$H_{2}+{\mathrm{OH}}^{-}+M⇄M-H_{\mathrm{ads}}+H_{2}O+e^{-}$$
(4)$$M-H_{\mathrm{ads}}+{\mathrm{OH}}^{-}⇄M+H_{2}O+e^{-}$$

Different strategies have been developed in order to enhance the hydrogen desorption from the active sites of metals and to increase the catalytic activity of the materials. Metallic alloys of Pt and other less expensive metals are the best option to increase the catalytic activity toward the HOR, since it is hypothesized that the oxophylic nature of a less noble metal can provide adsorbed OH species, facilitating the desorption of adsorbed hydrogen (accordingly to Heyrovsky and Volmer steps) and, simultaneously, inducing electronic effects, decreasing the bonding energy of M–H [24,25,26].

The main goal of this work was to determine the best carbon support for a Pt-based catalyst, among carbon nanotubes (MWCNTs), reduced graphene oxide (rGO), and carbon. The best performance support was used for the deposition of Co nanoparticles in order to obtain a template by the direct chemical reduction method. To use optimal Pt loading, this metal is deposited by a galvanic displacement method. The performance and changes in the mechanism of the HOR are evaluated for all the bimetallic catalysts to correlate the Pt loading and the understanding of the HOR. Optimum Pt loading is an essential synthesis parameter in order to obtain low-cost/high-performance anodic catalysts for the AEMFC and the HP.

## 2. Materials and Methods

### 2.1. Reagents

Sulfuric acid (H_2_SO_4_, 98%), nitric acid (HNO_3_, 70%), toluene (C_7_H_8_, 99.5%), ethylenediamine (C_2_H_8_N_2_, ˃99%), acetone (C_3_H_6_O, 99.9%), methanol (C_2_H_4_O, 99.9%), and ethanol anhydrous (CH_3_CH_2_OH, 99.9%) were acquired from Fisher Scientific. Sodium borohydride (NaBH_4_, ≥96%), ferrocene (C_10_H_10_Fe, 98%), Nafion^®^ 5% *v*/*v* solution (C_7_HF_13_O_5_S·C_2_F_4_, 95%), cobalt chloride (II) (CoCl_2_·2H_2_O), and potassium hexachloroplatinate (IV) (K_2_PtCl_6_, 99+%) were supplied by Sigma Aldrich (St. Louis, MO, US). Hydrogen ultra-high-purity grade (99.999%), argon ultra-high-purity grade (99.999%), nitrogen industrial-grade purity (99.985%), and carbon monoxide compressed (99.9%) were purchased from Infra. All solutions were prepared with deionized water (milliQ grade, 18 MΩ).

### 2.2. Synthesis of the Materials

#### 2.2.1. Synthesis of the Support (MWCNT and rGO)

MWCNTs were synthesized according to the procedure reported by Aguilar-Elguézabal et al., with minor modifications, where toluene (C_7_H_8_, 99.5%) was used as the carbon source instead of benzene [27]. The collected MWCNTs were oxidized by dispersion in an acidic solution (H_2_SO_4_:HNO_3_ 1:3 *v*/*v*) at reflux for 120 min in order to generate oxygen surface groups that act as active sites for the deposition of metallic nanoparticles [28,29]. rGO was synthesized according to the modified Hummer method [30].

#### 2.2.2. Synthesis of the Pt-Based Catalyst (Pt/MWCNT, Pt/rGO, and Pt/C)

Pt/MWCNT and Pt/rGO catalysts were synthesized by the method described in our previous reports [31,32]. Commercial Pt/C (20% of Pt loading) was purchased from Fuel Cell Store^®^ (College Station, TX, USA) and used as received.

#### 2.2.3. Synthesis of the Template Co/MWCNT

The Co/MWCNT template was synthesized by the direct chemical reduction of the metallic complex; after dissolving 80 mg of the oxidized MWCNTs in 200 mL of methanol, the mixture was dispersed by sonication for 30 min. Meanwhile, 540 mg of CoCl_2_·2H_2_O was dissolved in 20 mL of methanol and 600 μL of ethylenediamine was added to obtain the metallic complex. Afterward, the metallic complex solution was added to the MWCNT dispersion. Then, a solution containing 960 mg of NaBH_4_ in 20 mL of methanol was added to the dispersion and allowed to react for 240 min under reflux. Finally, the solution was filtered and the remaining solid washed with DI water, methanol, and acetone and dried in an oven for 24 h.

#### 2.2.4. Synthesis of the Catalysts PtCo/MWCNT

The galvanic displacement method consists in the deposition of a noble metal over an interface of another metal, promoted by the standard potential of the involved redox species. The displacement reaction proceeds in a spontaneous manner, where the more noble metal (more positive E^0^) in the solution is deposited, while the less noble metal (less positive E^0^) is dissolved in the solution. The morphology, structure, and amount of the noble metal deposited depend upon several parameters, such as the ΔE^0^ of the redox species and physicochemical conditions of the media. Nevertheless, the morphology and structure of the composite can be influenced by the reaction conditions. Under this consideration, the ultrasound-assisted galvanic displacement method provides sonochemical energy to enhance the dispersion of the metallic nanoparticles [33,34,35]. Since the main objective of this work is to synthesize low-Pt-loading-based catalysts, the ultrasound-assisted galvanic displacement method is proposed to obtain PtCo/MWCNT, according to the following procedure.

A solution of 20 mg of the Co/MWCNT template in 25 mL of DI water was dispersed in a beaker under sonication. Meanwhile, different aqueous solutions of K_2_PtCl_6_ were prepared, taking the corresponding volume of a 200 ppm K_2_PtCl_6_ aqueous solution, diluted to 25 mL in a volumetric flask. Both solutions were mixed, and galvanic displacement of the Co by the Pt ions was carried out assisted by an ultrasonic probe (Sonics, model Vibra cell 750) at 30% of the maximum intensity (maximum power intensity 750 W), at 60 Hz for 5 min, applying a 1:1 on/off pulse. The Pt loading deposited is controlled by the volume of the K_2_PtCl_6_ added to the solution. Table 1 shows the volume of the K_2_PtCl_6_ added for each material.

After the galvanic displacement method, the solid was washed with DI water and dried in an oven for 24 h.

### 2.3. Physicochemical Characterization

Scanning electron micrographs were taken by using a Tescan Vega 3 (Brno, Czech Republic) operated at 25 kV with a secondary electron detector, and elemental analysis was performed by EDS analysis with Bruker XFlash Detector (Billerica, MA, US).

Transmission electronic micrographs were performed on a Jeol JEM 2200FS (Tokyo, Japan) microscope operated at 200 kV. The microscope was used with a spherical aberration corrector in a probe mode. The bimetallic nature of the sample was determined by acquiring images in STEM mode using a high-angle annular dark field (HAADF) detector to compare Z contrast images.

The crystallite parameters and crystal phase identification of the Pt Co catalysts were obtained by X-ray diffraction analysis with Bruker D8 Advance (Billerica, MA, USA).

Thermogravimetric analyses were recorded in a TA instrument Q600 in order to determine the metallic loading of the catalysts. The Pt and Co loading was calculated by inductively coupled plasma–optic emission spectroscopy (ICP-OES) performed in Perkin Elmer Optima 8300 (Waltham, MA, US). The samples were thermally treated overnight in an oven at 800 °C. The solids obtained from each sample were dissolved in an acidic solution (HNO_3_:HCl 1:3 *v*/*v*). The aliquots were analyzed in axial mode under gas, auxiliary, and nebulizer flows of 15.00, 0.20, and 0.55 L min^−1^, respectively. The radio frequency power was set up at 1300 W, and the volume of the aliquots was 1 mL per analysis. The wavelengths for Pt and Co were 265.95 and 228.62 nm, respectively. To ensure accuracy, all the samples were analyzed three times. Table 1 shows the percentages of Pt and Co calculated on the basis of TGA and ICP-OES analysis of each sample.

### 2.4. Electrochemical Evaluation of the Catalysts

#### 2.4.1. Electrochemical Cell and Instrument for the Determination of the Catalytic Activity

A typical three-electrode cell was used to study the HOR mechanism and to evaluate the performance of the catalyst in the reaction. A glassy carbon rotating disk electrode (GCRDE, 0.2 cm^2^ of the geometric area) was used as the working electrode. A Pt coil was employed as the auxiliary electrode, and Hg/HgO/KOH 1 M was used as the reference electrode. The working electrode was attached to a rotation rate control unit (Pine Research MSR model (Durham, NC, USA)). The electrochemical tests were performed with a potentiostat/galvanostat bioanalytical system model Epsilon. The composition of the catalytic inks prepared was 2 mg of the catalysts, 150 μL of the Nafion^®^ solution, and 550 μL of ethanol. The suspension was dispersed under sonication for 30 min, prior to modification with the catalytic inks. The GCRDE was polished with an alumina slurry (0.05 μm); then 40 μL of the ink was carefully deposited onto the GCRDE and allowed to dry for 10 min or until a thin homogeneous layer was obtained. Before each electrochemical measurement, the working electrode was activated by imposing 30 cyclic voltammograms in the potential range of −1.1 to 0.7 V vs. Hg/HgO/KOH 1 M in NaOH 0.1 M under a saturated nitrogen atmosphere at a scan rate of 100 mV s^−1^ in order to eliminate adsorbed species in the catalytic sites. Afterward, linear sweep voltammograms were recorded from −1.0 to 0.7 V vs. Hg/HgO/KOH 1 M in NaOH 0.1 M under a saturated hydrogen atmosphere at a scan rate of 10 mV s^−1^. For all the catalysts, the linear sweep voltammograms were executed in a wide laminar regimen range from 0 to 1600 rpm. All potentials in this work are referred against a reversible hydrogen electrode.

#### 2.4.2. Estimation of the Electrochemically Surface Area

The electrochemical surface area (ECSA) was measured by a CO stripping test. A routine was imposed with the aim to simulate the electrochemically induced adsorption/desorption of carbonaceous species at the active sites. The electrochemical system was similar to the system described in the catalytic activity section. Only the reference electrode employed was Ag/AgCl/KCl 3 M, since the studies for the ECSA estimation was performed in acidic media. Firstly, 30 cyclic voltammograms were recorded from −0.2 to 1.0 V vs. Ag/AgCl/KCl 3 M in H_2_SO_4_ 0.5 M at a scan rate of 100 mV s^−1^. Then, a potentiostatic pulse of −120 mV vs. Ag/AgCl/KCl 3 M was imposed for 3 min in the H_2_SO_4_ 0.5 M solution saturated with carbon monoxide, allowing the adsorption of the CO species. Finally, 3 cyclic voltammograms were recorded from −0.2 to 1.0 V vs. Ag/AgCl/KCl 3 M in H_2_SO_4_ 0.5 M at a scan rate of 100 mV s^−1^, showing an intense peak located at 0.7 V vs. Ag/AgCl. This might be attributed to the desorption of the CO species [36,37]. From the integration of this peak, the ECSA is calculated according to Equation (5):(5)$$\mathrm{ECSA}=\frac{Q}{Q_{\mathrm{CO}}L_{Pt}}$$
where the ECSA is measured in m^2^ g^−1^_Pt_, *Q* is the integrated charge in mC, Q_CO_ is the constant for the required charge for the desorption of a monolayer of carbon monoxide (0.42 mC cm^−2^), and *L_Pt_* is the Pt loading estimated from TGA and ICP-OES analysis.

## 3. Results and Discussion

### 3.1. Physicochemical Characterization

Figure 1a shows XRD patterns of synthesized PtCo/MWCNT catalysts. The template Co/MWCNT exhibited two diffracted intensity peaks, at 26.4° (002) and 43.1° (100), that indexed to the structure of graphite reflections (JCPDS card No. 96-120-0018), which have a decrease in the crystallinity order of MWCNTs due to functionalization treatment. In addition, the diffracted intensity peaks at 34.5° (111), 42.6° (200), and 56.8° (220) matched the pure structure of CoO (JCPDS card no. 00-042-1300; space group F-4_3m_ (216), cubic structure, a = 0.45400 nm) and 36.4° (311), 30.9° (202), and (511) indexed to the pure structure of Co_3_O_4_ (Crystallography Open Database (COD) card no. 96-900-5895, space group F d-3m (227) cubic, a = 0.818930 nm). It is important to point out that Co/MWCNT exhibited broad and diminished peaks, attributed to the amorphous structure obtained by the synthesis method. The hybrid materials with Pt NPs (i.e., from Pt Co2 to Pt Co6) also showed two diffracted intensity peaks, at 33.6° (110) and 59.2° (211), that can be attributed to the structure of Co_0.5_Pt_0.5_ (COD card no. 96-152-4153, space group F d-3m (227) cubic, a = 0.818930 nm). It is noticeable that the peaks at 34.5°, 56.8°, 30.9°, and 36.4° decrease in intensity when the Pt concentration increases. This fact indicates the hindering effect of the Co nanoparticles by the Pt nanoparticles. Simultaneously, the peak at 33.6 increases due to the increasing Pt concentration. Since the deposition of Pt by the galvanic displacement method is randomly dispersed, the deposition of Pt takes place on Co without distinction of the crystallite structure or type of Co oxide. The evidence of this fact is the well-dispersed HRTEM images (Figure 1d–f).

Conversely, the three diffracted intensity peaks at 40.0, 46.8, and 67.0 of 2θ related to the cubic close-packed structure of Pt were imperceptible. The result suggests that functionalized MWCNTs were covered with a mixture of cobalt oxides (i.e., CoO and Co_3_O_4_) where the galvanic displacement method used allowed the deposition of Pt on the sites of the cobalt oxide surface. SEM images show the tubular structure of the MWCNT and particle agglomeration in a few zones (Appendix A). However, EDS mapping analysis (Appendix A) shows that Pt and Co nanoparticles were dispersed along the MWCNT in a narrow distribution, as is indicated in Figure 1b,c. Nevertheless, the particle size of the particles cannot be determined by SEM due to the resolution of the instrument.

Figure 1d–f shows HRTEM images of the hybrid materials. TEM images show Co NPs of 2 to 20 nm coating the MWCNT surface, forming a wrap in several regions of the carbon support (Figure 1d). In addition, the growth of Pt NPs on the surface of Co NPs occurs by the galvanic displacement method (Figure 1e,f). Co and Pt NPs can be distinguished by an HAADF detector. Since this detector is sensitive to the atomic numbers of certain elements, hence the higher the atomic number, the brighter the appearance in the TEM image.

Table 1 summarizes the shape parameters of PtCo/MWCNT catalysts. The particle diameter (D_p_) and area particle (A_p_) were similar for all catalysts. Figure 1b shows the Gaussian probability density curves for D_p_. Moreover, Figure 1c shows the log-normal probability density curves for A_p_. The graphs confirm the homogeneity of the distribution of Pt NPs on the Co NP surface, indicating the reproducibility of the synthesis method in terms of the growth and size of Pt NPs. Appendix A shows HRTEM images of Co NPs covered with Pt NPs for each catalyst. Interestingly, the number of Pt NPs on Co NPs was observed to change according to the treatment, showing an increase with an increase in the Pt concentration. The ratio between Pt NPs A_p_ (nm^2^) and Co NPs A_p_ (nm^2^) is also shown in Table 1.

The observed difference between the number of Pt NPs and the covered area of Co NPs is significant. The increase in the active sites of Pt is related to a better performance of the catalyst. However, a higher number of Pt NPs increases the presence of agglomerations, decreasing the inter-particle distance. This condition changes the diffusion conversion from spherical to planar, associated with the active area decrease because of the loss of available area, modifying the transport of materials, leading to a low performance of the catalyst.

Appendix A provides SEM and TEM micrographs of PtCo-based catalysts, where it is important to point out that the diameter of the MWCNTs is 60 nm approximately (Appendix A). Moreover, it is noticeable that metallic nanoparticles are well dispersed along the MWCNTs due to the random distribution of the metallic particles exhibited by the EDS mapping analysis. It is possible to distinguish Pt from Co nanoparticles since the smaller Pt nanoparticles appear brighter than Co nanoparticles in the HRTEM (Appendix A). The size distribution of Co and Pt nanoparticles at the bimetallic PtCo/MWCNT 4 catalyst were 8.3 ± 1.7 nm and 1.25 ± 0.3 nm for the Co and Pt nanoparticles, respectively (Appendix A). The same size distribution trend was observed for all bimetallic PtCo/MWCNTs independently of the K_2_PtCl_6_ solution added, demonstrating that the galvanic displacement method has no influence on the NP size. Conversely, the Pt surface on the support increases linearly with the volume of K_2_PtCl_6_ added (not shown).

Figure 2a shows the thermogravimetric analyses of the catalysts of Pt on different supports, where the Pt loading was similar in the three catalysts. The Pt/MWCNT catalyst exhibited the first decomposition step between 200 and 300 °C, which is associated with the oxygen-adsorbed species on the surface of the MWCNT. The other important decomposition step appears at 550–700 °C, which is related to the oxidation of the carbon support. Figure 2b shows the thermogravimetric analyses of the PtCo/MWCNT. All the catalysts exhibited almost the same thermal behavior. However, PtCo/MWCNT catalysts showed lower thermal stability. Since these catalysts exhibit better dispersion than the template, the well-dispersed nanoparticles accelerate the corrosion rate of the support [38]. The residual weight, associated with the most stable oxides of the metals, is inset in Figure 2b.

ICP-OES was performed in order to elucidate the Pt and Co loadings in the catalysts. Table 2 summarizes the Pt and Co loadings of bimetallic catalysts, where it is noteworthy that the Pt loading for all catalysts is below 4%, which represents progress regarding the commercial Pt/C (20%). It is important to mention that there is a trend where the percentage of Pt increases when more K_2_PtCl_6_ solution is added and the percentage of Co decreases as a side effect of the galvanic displacement phenomena. Pt loading is a critical aspect in anodic catalysts for AEMFC applications, since the economics and the performance exhibit an opposite relationship. Hence, the mechanism and performance of the catalysts in the HOR should be discussed before the selection of a potential anodic catalyst for AEMFC application.

### 3.2. Active Surface Area (ECSA) and the Oxophylic Nature of PtCo Catalysts

Figure 3a shows the cyclic voltammograms of the CO stripping for all catalysts. Co/MWCNT and PtCo/MWCNT 1 do not exhibit any feature for CO oxidation, which implies that the ECSA for these materials was not computed (inset of Figure 3).

PtCo/MWCNT bimetallic catalysts, Pt/C, and Pt/MWCNT exhibited intense peaks between 0.65 and 0.75 V vs. Ag/AgCl, attributed to CO desorption from the active sites. It is noticeable that increasing the Pt loading decreases the potential peak related to CO stripping, which means a lower activation energy for CO conversion. A close relationship between Pt loading and the current density is expected. However, an inconsistency is found for PtCo/MWCNT 3 since this catalyst exhibits a higher current density than PtCo/MWCNT 4, which is attributed to a better dispersion of the Pt active sites. Table 3 summarizes the potential peak, the current density, the integrated charge, and the ECSA for each catalyst.

Figure 3b shows the cyclic voltammograms in the pure electrolyte solution, exhibiting the adsorption (anodic scan) and desorption (cathodic scan) of the oxygen species on the surface of the catalysts. Mei and coworkers reported previously the anodic peaks for the Co(II) toward Co(III) and Co(III) toward Co(IV) at 300 and 600 mV, respectively [38]. Hence, there are two cathodic peaks for the desorption of the oxygen species at the catalyst surface. The Co/MWCNT exhibited broad peaks at 345 and 695 mV.

The incorporation of Pt nanoparticle onto the surface of the catalysts promotes the displacement of the cathodic peaks toward more negative potentials, indicating more energy for the desorption of the oxygen species. This phenomenon is associated with the oxophylic nature of the Co-based materials. All the PtCo catalysts showed the same displacement for both cathodic peaks, establishing the decreasing order PtCo/MWCNT 4 > PtCo/MWCNT 6 > PtCo/MWCNT 2 > PtCo/MWCNT 3 > PtCo/MWCNT 1 for the availability of the OH-adsorbed species. The oxophylic nature is a critical aspect of catalysts where the Heyrovsky and/or Volmer is the RDS in the HOR (Tafel slope value around 120 mV dec^−1^), since in those steps, the limiting conditions are the electronic charge transference and/or the OH-adsorbed species.

### 3.3. HOR Catalytic Activity and Mechanism

The catalytic activity in an electrochemical process is usually referred to in terms of the potential (related to activation energy) and the current density (related to kinetic rate of the reaction). To determine the best carbonaceous support, the catalytic activity for the HOR of Pt-supported catalysts is evaluated. As per Figure 4a, the linear voltammograms of all the Pt supported on carbonaceous structures exhibits three typical regions of the limiting mass transport system: the kinetic region (E < 50 mV), where the density current increases with the potential; the mass transport region (E > 150 mV), where the current density is independent of the potential but varies with the rotation rate of the electrode (besides, a plateau is reached in this region); and the mixed region (50 mV < E < 150 mV), where the current density varies according to the potential and rotation rate of the electrode. Different kinetics features can be determined from these regions. Half-wave potential (E_1/2_) is founded in the mixed region, and the limiting current density (J_lim_) appears at the mass transport region. Pt/MWCNT exhibited the lowest E_1/2_ and the highest J_lim_ values among different carbonaceous supports, which are related to the lowest activation energy for the HOR and the highest kinetic rate and/or more active sites available for the catalysis of the HOR, respectively. The inset in Figure 4a exhibits the impedance spectra of the Pt catalysts on different carbonaceous supports, where the semi-circle in the high-frequency range is related to the resistance to the transference charge (R_tc_). The trend for the R_tc_ in the carbonaceous supports was Pt/MWCNT < Pt/C < Pt < rGO, which is consistent with the trend for the catalytic activity. Hence, MWCNTs are selected as the best support for HOR application.

Figure 4b shows the linear voltammograms of the template and the bimetallic catalysts. The Co/MWCNT does not exhibit the plateau attributed to hydrogen oxidation under a hydrodynamic regime. Hence, this material does not catalyze the HOR. The broad oxidation peak centered at 0.82 V is related to Co oxidation from Co(II) to Co(III) oxidation state [39,40]. Similarly, PtCo/MWCNT 1 exhibited a shift in the potential peak of the same signal toward a more negative potential, from which, it can be stated that there is a critical amount of Pt loading required to catalyze the HOR.

Another key aspect to mention is that except for PtCo/MWCNT 1, the other PtCo/MWCNT catalysts exhibited the plateau as well, supporting the hypothesis that there is a critical value of Pt loading necessary for the catalysis of the HOR. However, regarding Pt/MWCNT, the bimetallic catalysts exhibited an E_1/2_ shift toward more positive potentials, which is related to a larger activation energy than Pt/MWCNT. There is a noticeable trend in the E_1/2_ of bimetallic catalysts and Pt loading, where it is assumed that increasing the Pt loading lowers the activation energy required for the HOR. The incorporation of Co nanoparticles in the bimetallic catalyst approach is carried out with the aim to decrease the cost and provide OH-adsorbed species, which can facilitate HOR catalysis if the Heyrovsky step or the Volmer step is the RDS of the HOR (a Tafel slope of about 120 mV dec^−1^). Although Co nanoparticles provide an oxophylic character to the catalysts and decrease the M–H binding energy, the evidence of increasing E_1/2_ suggests that OH^−^ ions do not limit the HOR. Hence, (i) the Tafel step (Equation (2)) reveals the RDS of the HOR or (ii) there is a complex HOR process where two different steps control the kinetic rate of the HOR. Moreover, another trend is noticeable in the J_lim_, where increasing the Pt loading of the bimetallic catalysts increases the J_lim_. Nevertheless, an inconsistency is found for PtCo/MWCNT 6, which might be attributed to a poor dispersion of the Pt nanoparticles on the surface support, decreasing the active sites. The best bimetallic catalyst is not identified only on the basis of catalytic activity evidence. Deeper insights are needed in order to clarify the HOR mechanism in all catalysts.

Regarding the reproducibility of the HOR experiments, linear voltammograms for PtCo/MWCNT 4 and PtCo/MWCNT 6 were performed in triplicate, where the relative standard deviation (RSD) values for Jl_im_ were ±6.22% and ±3.30% for PtCo/MWCNT 4 and PtCo/MWCNT 6, respectively. Similarly, the RSD values for E_1/2_ were ±4.11% and ±5.24% for PtCo/MWCNT 4 and PtCo/MWCNT 6, respectively. These values suggest a high reproducibility of the catalytic activity parameters of the catalysts.

Figure 5a shows the Koutecky–Levich analysis for bimetallic catalysts. Pt/C and Pt/MWCNT are included for comparison purpose. It is noteworthy that all the PtCo/MWCNT bimetallic catalysts and Pt/MWCNT exhibited a K–L coefficient value between 3.16 and 3.71 cm^2^ mA^−1^ s^−1/2^, which is relatively close to the theoretical value reported by Zhuang and coworkers of 4.87 cm^2^ mA^−1^ s^−1/2^ for the Pt polycrystalline, leading to the conclusion that the HOR mechanism in these catalysts is via the transfer of two electrons.

Figure 5b shows the Tafel analysis of the catalysts, where Pt/C and Pt/MWCNT reach values around 120 mV dec^−1^, indicating that either the Heyrovsky step or the Volmer step (electronic charge transference) is the RDS of the HOR mechanism. This is also consistent with the poor oxophylic feature of the Pt particles. Contrary to expectation, PtCo/MWCNT 4 and PtCo/MWCNT 6 exhibited Tafel slope values of 120 mV dec^−1^, even with the presence of Co nanoparticles, which can be attributed to the electronic charge transference associated with the Heyrovsky or Volmer step and a high activation energy (compared to Pt/C and Pt/MWCNT) as a side effect. Conversely, PtCo/MWCNT 2 and PtCo/MWCNT 3 exhibited intermediate Tafel slope values, which is related to changes in the RDS of the mechanism. Since a Tafel slope value of 30 mV dec^−1^ is associated with a dissociative hydrogen adsorption on Pt active sites, a mixed Tafel step is the RDS on the HOR mechanism for low-Pt-loading catalysts (1.5% < Pt loading < 2.7%), which is consistent with the fact that OH-adsorbed species on Co sites do not limit the reaction; rather, the active metal sites are the unique species that are involved in the RDS.

Table 4 summarizes the kinetic parameters of the catalysts from K–L and Tafel analyses.

### 3.4. Stability of the Catalysts

To determine the best catalyst for the HOR, a series of potentiodynamic cycles were imposed in the modified electrodes with PtCo/MWCNT 4 and PtCo/MWCNT 6. The catalytic activity parameters (J_lim_ and E_1/2_) were measured after the potentiodynamic cycles and compared with their respective initial values. The potentiodynamic cycles were recorded from −1.0 to 1.0 V vs. Hg/HgO/KOH 1.0 M at 100 mV s^−1^. This wide potential range ensures that hydrogen/oxygen evolution reactions take place at the cathodic/anodic scan; hence the material is subjected to harsh conditions. Figure 6a,b shows the stability of PtCo/MWCNT 6 and PtCo/MWCNT 4, respectively, on the basis of the catalytic activity parameters.

Both catalysts exhibited an increase in the values of E_1/2_ and J_lim_, which is attributed to the activation of Pt sites. However, after 1500 cycles, the PtCo/MWCNT 4 reaches the initial values of the catalytic activity parameters but PtCo/MWCNT 6 reaches the initial values of the aforementioned parameters, which implies a higher stability for this catalyst. Finally, after 3000 cycles, PtCo/MWCNT 4 is left with 61% of the E_1/2_ initial value and 27% of the J_lim_ initial value, while PtCo/MWCNT 6 is left with 88% of the E_1/2_ initial value and 54% of the J_lim_ initial value.

## 4. Conclusions

The performance and mechanism of the HOR for PtCo/MWCNT catalysts were studied in order to optimize the Pt loading on the catalyst. From all carbonaceous supports, Pt/MWCNT exhibited the lowest E_1/2_ and the highest J_lim_, which is related to low activation energy and a high kinetic rate constant or a large number of active sites available, respectively.

To decrease the Pt loading, five bimetallic PtCo/MWCNT were synthesized. The catalyst with the minimum Pt loading (PtCo/MWCNT 1, Pt loading around 0.4%) did not exhibit a response toward the HOR or the Co/MWCNT template. From the bimetallic catalysts that exhibited catalytic activity, the PtCo/MWCNT with the higher Pt loading (PtCo/MWCNT 4 and PtCo/MWCNT 6, Pt loading of 3.1% and 4.0%, respectively) exhibited the best catalytic activity and a mechanism similar to that of pure Pt/MWCNT, even with lower Pt loading (decreased by 5 times) than the value of the commercial Pt/C catalyst, which decreases significantly the cost of the anodic catalyst for AEMFC.

The oxophylic nature of the Co nanoparticle provided OH-adsorbed species, enabling the study of the RDS at different Pt loading values. Since the Tafel slope increases with higher Pt loading, changes in the HOR mechanism were observed. High-Pt-loading catalysts showed electron charge transference (Heyrovsky or Volmer step) as the RDS, while low-Pt-loading catalysts (PtCo/MWCNT 2 and PtCo/MWCNT 3, with Pt loading of 1.6% and 2.7%, respectively) exhibited intermediate Tafel slope values (leading to the Tafel step as the RDS), where the dissociative hydrogen adsorption is the RDS, since low-Pt-active sites limit the HOR irrespective of the large number of OH-adsorbed species in the Co sites.

The stability performance of the PtCo catalysts was evaluated (PtCo/MWCNT 6 and PtCo/MWCNT 4), where an increase in the E_1/2_ and J_lim_ initial values was observed. PtCo/MWCNT 4 decreases the catalytic activity parameters in fewer potentiodynamic cycles than PtCo/MWCNT 6. Moreover, PtCo/MWCNT 6 has higher values after 3000 potentiodynamic cycles, which is associated with higher stability.

The findings in this study are a guidance for the design and preparation of affordable and high-performance anodic catalysts for AEMFC and HP applications.

## Figures and Tables

**Figure 1 nanomaterials-11-03156-f001:**
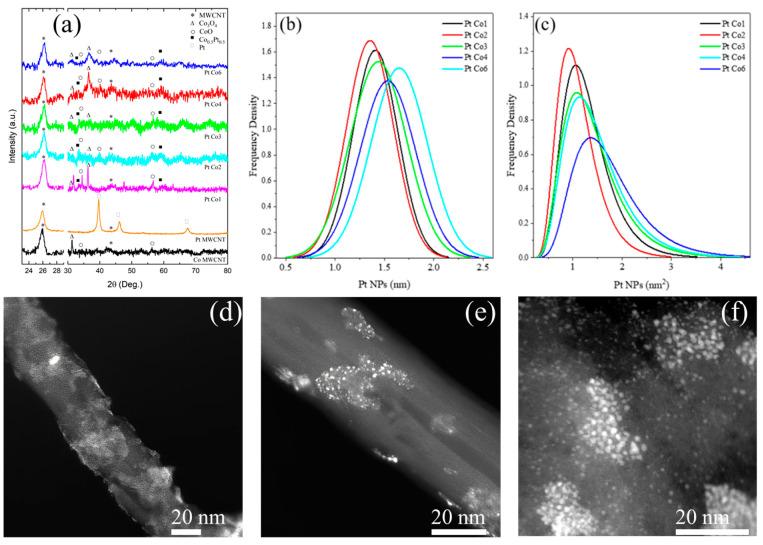
(**a**) XRD analysis, (**b**) Pt size distribution, (**c**) Pt NP area distribution, and (**d**–**f**) HRTEM images of the PtCo/MWCNT 4 catalyst.

**Figure 2 nanomaterials-11-03156-f002:**
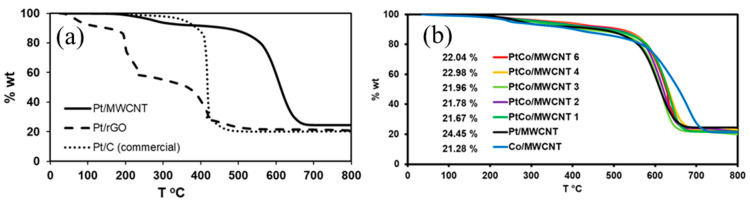
(**a**) Thermogravimetric curves of Pt/MWCNT, Pt/C, and Pt/rGO under air atmosphere. (**b**) Thermogravimetric curves of the PtCo/MWCNT catalysts under air atmosphere; the metal loading mass is inset in the figure.

**Figure 3 nanomaterials-11-03156-f003:**
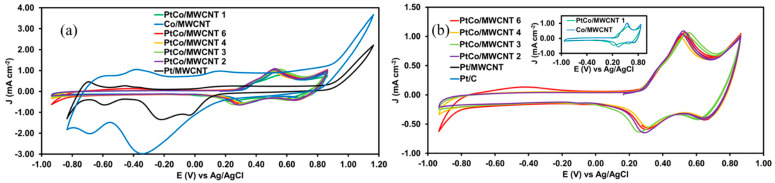
(**a**) Cyclic voltammograms of PtCo/MWCNT catalysts after the application of a potentiostatic pulse of −120 mV for 3 min under a CO atmosphere. (**b**) Cyclic voltammograms of PtCo/MWCNT catalysts in the pure electrolyte solution. The solution was H_2_SO_4_ 0.5 M at a scan rate of 100 mV s^−1^.

**Figure 4 nanomaterials-11-03156-f004:**
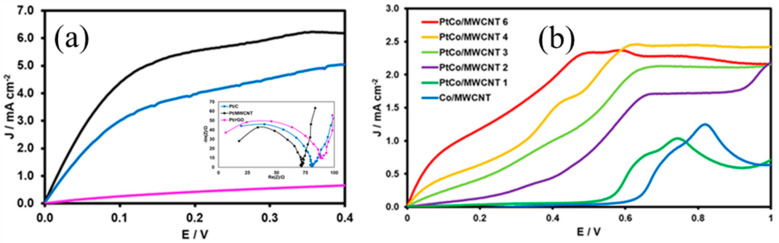
Linear sweep voltammograms of the (**a**) Pt catalysts on different carbonaceous supports (inset: electrochemical impedance spectra of the Pt catalysts in different supports) and (**b**) PtCo/MWCNT catalysts with different Pt loading in a solution of NaOH 0.1 M saturated in hydrogen at 1600 rpm at a scan rate of 10 mV s^−1^.

**Figure 5 nanomaterials-11-03156-f005:**
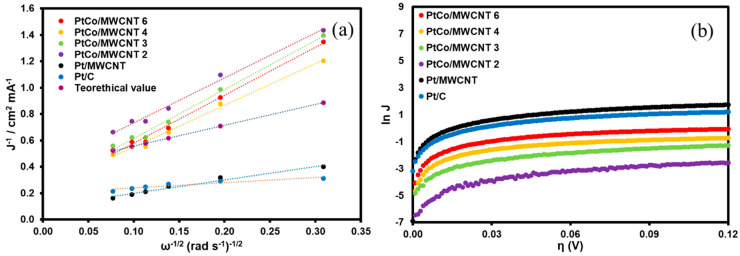
(**a**) Koutecky–Levich and (**b**) Tafel plots for PtCo/MWCNT; for comparison purpose, Pt/C and Pt/MWCNT are included in both analyses.

**Figure 6 nanomaterials-11-03156-f006:**
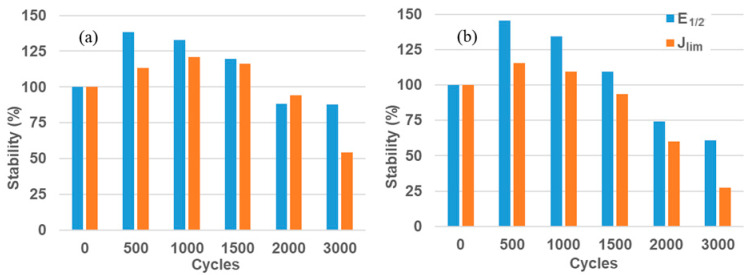
Evolution of the stability of the catalytic activity parameters (E_1/2_ and J_lim_) of (**a**) PtCo/MWCNT 6 and (**b**) PtCo/MWCNT 4.

**Table 1 nanomaterials-11-03156-t001:** Morphological parameters and crystallite size of the PtCo catalysts.

Catalysts	^†‡^ Pt NPs D_p_ (nm)	^†‡^ Pt NPs Ap (nm^2^)	Pt NPs Ap/Co NPs A (%)
PtCo/MWCNT 1	1.41 ± 0.23	1.30 ± 0.51	6.2
PtCo/MWCNT 2	1.35 ± 0.24	1.11 ± 0.41	6.6
PtCo/MWCNT 3	1.44 ± 0.28	1.41 ± 0.62	16.1
PtCo/MWCNT 4	1.53 ± 0.28	1.72 ± 0.65	20.2
PtCo/MWCNT 6	1.65 ± 0.29	1.47 ± 0.60	25.4

^†^ The number of particles was higher than 300 in the system. ^‡^ Pt NPs Dp exhibited normal density function (*p*-value > 0.05), and Pt NPs Ap have log-normal density function (*p*-value > 0.05).

**Table 2 nanomaterials-11-03156-t002:** Pt concentration solution for each catalyst and metal percentages (Pt and Co) of the synthesized catalysts.

Catalysts	K_2_PtCl_6_ Solution Added (mL)	Pt Added (μmol_Pt_)	Pt Concentration (μM)	%Pt ^1^	%Co ^1^
PtCo/MWCNT 1	1	1.026	20.513	0.438	13.117
PtCo/MWCNT 2	2	2.051	41.026	1.568	12.176
PtCo/MWCNT 3	3	3.077	61.538	2.709	12.065
PtCo/MWCNT 4	4	4.102	82.051	3.108	11.718
PtCo/MWCNT 6	6	6.154	123.077	4.034	10.134

^1^ Calculated on the basis of ICP-OES and TGA.

**Table 3 nanomaterials-11-03156-t003:** Summary of the CO oxidation peak (E_p_), CO current density peak (J_p_), integrated electric charge (Q), and active surface area (ECSA) for the catalysts of this work.

Catalysts	E_p_ (mV vs. Ag/AgCl)	J_p_ (mA cm^−2^)	Q (mC)	ECSA (m^2^ gr^−1^_Pt_)
Pt/C	663	21.40	7.17	74.7
Pt/MWCNT	665	14.50	2.57	27.72
Co/MWCNT	-	-	-	-
PtCo/MWCNT 1	-	-	-	-
PtCo/MWCNT 2	750	2.74	0.43	56.70
PtCo/MWCNT 3	720	8.16	0.83	63.56
PtCo/MWCNT 4	685	6.51	0.88	59.17
PtCo/MWCNT 6	680	9.70	1.44	74.48

**Table 4 nanomaterials-11-03156-t004:** Half-wave potential (E_1/2_), limiting current density (J_lim_), Koutecky–Levich slope, Tafel slope, electronic charge coefficient (α), and exchange current density (J_0_) for the catalysts of this work.

Catalysts	E_1/2_ (mV)	J_lim_ (mA cm^−2^)	K–L Slope (cm^−2^ mA^−1^ s^−1/2^)	Tafel Slope (mV dec^−1^)	A	J_0_ (mA cm^−2^)	Ref.
Pt/C	76	5.00	1.22	123.9	0.21	1.36	This work
Pt/MWCNT	60	6.20	3.16	121.8	0.21	2.21	This work
Pt/rGO	165	0.65	-	-	-	-	This work
Co/MWCNT	-	-	-	-	-	-	This work
PtCo/MWCNT 1	-	-	-	-	-	-	This work
PtCo/MWCNT 2	510	1.72	3.41	73.4	0.35	0.02	This work
PtCo/MWCNT 3	420	2.12	3.71	90.7	0.28	0.08	This work
PtCo/MWCNT 4	364	2.44	3.16	121.3	0.21	0.20	This work
PtCo/MWCNT 6	187	2.28	3.63	120.5	0.21	0.39	This work
Pt(pc)	65	2.70	4.87	125	0.50	-	[18]
Ni/N-NTC	25	1.30	5.21	-	0.45	0.03 ^3^	[21]
Ni/C	28	1.60 ^1^	3.50	-	-	-	[41]
RuPt/C	15	3.20 ^2^	-	35	-	1.42 ^3^	[22]
RuPd/C	95	3.30 ^3^	-	219	0.27	0.15 ^3^	[22]
Pd/MnO_2_	279	2.33	-	35	-	10.4 ^3^	[42]
PtRu	27	2.10 ^1^	-	-	-	4.00	[43]
Ni/SC	25	1.42 ^1^	-	-	0.54	0.04 ^3^	[44]
MoNi_4_	18	2.85	4.60	48.6	-	3.41	[45]
WNi_4_	23	2.50	4.32	-	-	1.87	[45]

^1^ J_lim_ reported in a polarization curve performed at 5 mV s^−1^. ^2^ J_lim_ reported in a polarization curve performed at 2500 rpm. ^3^ Exchange current density reported in a normalized ECSA (mA cm^−2^_Metal_).

## Data Availability

The data presented in this study are available on request from the corresponding authors.

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
