# Peer review of "Ultra-Low Pt Loading in PtCo Catalysts for the Hydrogen Oxidation Reaction: What Role Do Co Nanoparticles Play?"

_nanomaterials, 2021, doi:10.3390/nano11113156_

Round 1

Reviewer 1 Report

This work reports the synthesis, characterization and performance evaluation of PtCo/MWCNT electrocatalyst for HOR in alkaline media. The investigation of the nature of PtCo/C catalyst for HOR catalysis is important for the design and the preparation of high-performance anodic catalyst for AEMFC and HP applications. However, the relation between structure and HOR performance is not well addressed and some results lack evidence. Therefore, I suggest the manuscript be resubmitted after addressing the following comments.

  1. In line 70, ammonia is not suitable for use as a fuel in PEMFC.
  2. Authors should clarify the formation mechanism of CoO and Co3O4 in the catalyst.
  3. The XRD pattern of pure Pt/MWCNT should be given to confirm the PtCo alloy structure. The analysis of the possible alloy phase needs evidence.
  4. Authors carried out the particle size statistics of Pt and Co particles how is the distribution of PtCo alloy nanoparticles?
  5. Pt is an excellent catalyst for HOR, but the catalytic activity of Pt catalysts in different when carbon supports are different. ICP test should be conducted to determine the Pt loading on different carbon supports.
  6. The conductivity of carbon supports is suggested to be measured in order to comprehensively discuss the HOR catalytic activity of the catalysts.
  7. Evidence should be given to investigate the role of Co for HOR.

Author Response

This work reports the synthesis, characterization and performance evaluation of PtCo/MWCNT electrocatalyst for HOR in alkaline media. The investigation of the nature of PtCo/C catalyst for HOR catalysis is important for the design and the preparation of high-performance anodic catalyst for AEMFC and HP applications. However, the relation between structure and HOR performance is not well addressed and some results lack evidence. Therefore, I suggest the manuscript be resubmitted after addressing the following comments.

The reviewer performed a deep critical analysis of our manuscript, in order to attend the observations from the reviewer point by point, authors response is highlighted in red: 

  1. In line 70, ammonia is not suitable for use as a fuel in PEMFC.

The sentence was clarified, to state that ammonia fuel cell is exclusively for AEMFC technology.

      2. Authors should clarify the formation mechanism of CoO and Co3O4 in the catalyst.

XRD patterns were performed again, the patterns exhibited an interestingly information about the structure and the phase of the composite PtCo/MWCNT nanomaterials.

First, the peaks located at 34.5°, 56.8° and 69.0° are related with the formation of CoO, where it shows that the peaks at 34.5° and 56.8° decrease their intensities, this is attributed to the hindering by the Pt nanoparticles of the CoO. On the other hand, the peaks at 30.9° and 36.4° are representative of the Co3O4, where the intensity of both peaks decrease as Pt concentration increases, which is in the same trend than CoO. This explanation was enhanced in the manuscript from line 266 to line 277.     

      3. The XRD pattern of pure Pt/MWCNT should be given to confirm the PtCo alloy structure. The analysis of the possible alloy phase needs evidence.

The XRD pattern of Pt/MWCNT was included in order to assign the Pt peaks, the XRD pattern of Pt/MWCNT is very clear, unfortunately the XRD patterns of the PtCo catalysts do not exhibited sharp/intense peaks, mainly attributed to the Pt loading lower than 4% and the high concentration of the Co (around 20%), which is deposited as amorphous phase. Moreover, other evidence of the PtCo alloy structure are the peaks at 33.6° and 59.2°, where the peak at 33.6° increases the intensity as increases the Pt concentration. Since the Pt concentration is too low in all the catalysts (lower than 4%), the expected peaks at 40.0°and 46.8° were imperceptibles in the PtCo catalysts, but well appreciated in Pt/MWCNT.

      4. Authors carried out the particle size statistics of Pt and Co particles how is the distribution of PtCo alloy nanoparticles?

How is stated in HRTEM micrographs, Figure 1(d)-Figure 1(f) exhibited a well dispersion of the Pt nanoparticles along the MWCNT, moreover, the EDS mapping analysis in the Figure SI 1 provide evidence about the Pt and Co randomly dispersed along the support. Only in few cases, Co nanoparticles exhibited agglomerations, attributed to the synthesis method. In the concentration range of the deposited Pt, there no was a significant effect of the dispersion on the particle size or Pt area, this discussion was already included in the manuscript in line 282 to line 293.

      5. Pt is an excellent catalyst for HOR, but the catalytic activity of Pt catalysts in different when carbon supports are different. ICP test should be conducted to determine the Pt loading on different carbon supports. 

This comment can take place to a confusion, since the theoretical Pt loading in Pt/MWCNT, Pt/rGO and Pt/C was around 20%, which was estimated by TGA, this figure was included in the manuscript as Figure 2(a). ICP-OES analyses were performed in order to elucidate the percentages of Pt and Co in bimetallic catalysts, ICP-OES does not estimate the Pt loading of the catalysts, since the samples of catalysts are digested and analyzed in liquid state. Catalysts with the same Pt loading in monometallic catalysts is not necessarily related with the Pt concentration, since the concentration is attributed to the mass of solid digested in the sample. Authors considered that is not necessary to explain this fact in the manuscript.

The changes in catalytic activity for Pt/MWCNT, Pt/rGO and Pt/C is attributed to the dispersion of the Pt nanoparticles, carbon conductivity and the availability of adsorbed OH groups to catalyze the HOR, in the order to determinate differences due to the support, Electrochemical impedance spectroscopy (EIS) tests were performed in order to evaluate the resistance to the transference charge, electrochemical impedance spectra was included as insert of Fig 4(a).

      6. The conductivity of carbon supports is suggested to be measured in order to comprehensively discuss the HOR catalytic activity of the catalysts.

Electrochemical impedance spectroscopy of the Pt deposited in carbonaceous supports was performed, this figure was included in the manuscript as inset of Figure 4(a). In this figure was observed a semi-circle at high frequency range related to the resistance to the transference charge, the trend for the carbonaceous supports was Pt/MWCNT<Pt/C<Pt<rGO, which is consistent with the trend for the catalytic activity. This discussion was included from line 419 to line 423. 

       7. Evidence should be given to investigate the role of Co for HOR.

The main evidence of the role of the Co nanoparticles is stated by Figure 3b, where the peak of OH adsorbed species located at -30 mV in Pt/MWCNT is displaced to more positive potentials (+345 mV), this fact indicates that OH bonding is weaker for PtCo catalysts than Pt/MWCNT, due to the oxophilic nature of Co nanoparticles. These displacements in the peaks explain the changes in the catalytic activity of the different PtCo catalysts. This discussion is already included in the manuscript (line 386 to line 391).

Authors would like to thank to the reviewer because take the opportunity to evaluate our manuscript. The authors believe that the addressed information will allow to improve the quality and the scope of the manuscript.

All the changes in the manuscript are highlighted in red, in order to make a difference with the first version of the manuscript.

Reviewer 2 Report

In this study, the authors tested several PtCo/carbonaceous supports for HOR activity with the main aim to economize Pt. In general, the article is interesting and can be considered for publication after a revision process.  

1) Synthesis process. Minor modifications during the MWCNTs synthesis must be disclosed for reproducibility purposes.  
2) What is the % fraction of CoO and Co3O4? 
3) HOR activity of the optimal composites vs. time (cycles) should be shown. 
4) It will be good to see XPS data of the optimal composites before and after HOR activity (after several cycles). 
5) Comment on the reproducibility, i.e. how many samples per trial were tested, HOR activity reproduction, etc.   

Author Response

In this study, the authors tested several PtCo/carbonaceous supports for HOR activity with the main aim to economize Pt. In general, the article is interesting and can be considered for publication after a revision process.  

The reviewer performed a breef summary of our work, in order to attend the observations from the reviewer point by point, authors response is highlighted in red:

1) Synthesis process. Minor modifications during the MWCNTs synthesis must be disclosed for reproducibility purposes.

Done, the minor modification compared to Ref 27 is the carbon source used in this manuscript, the correction was stated at line 141-142. 
2) What is the % fraction of CoO and Co3O4? 

The %CoO and %Co3O4 cannot be estimated by XRD, since the pattern does not exhibit intense peaks for neither the pure Co/MWCNT and the PtCo/MWCNT, this fact is attributed to the amorphous nature of Co deposited by the chemical reduction method, however, the presence of the signals in the patterns confirm both structures. An alternative method to estimate the presence of the CoO and Co3O4 is XPS, but authors do not have access to XPS instrument.

3) HOR activity of the optimal composites vs. time (cycles) should be shown. 

A series of potentiodynamic cycles were imposed for the best catalysts (PtCo/MWCNT 4 and PtCo/MWCNT 6) under harsh conditions, in order to evaluate the evolution of the catalytic activity parameters (E1/2 y Jlim), the potentyodinamic cycles were recorded with a wide potential range, where the reactions of the solvent take place (where the stability of the material is compromised by the hydrogen/oxygen evolution). The results are very interesting, since can determinate that PtCo/MWCNT 6 is the highest activity catalyst and the most endurable for potential AEMFC/HP applications. The discussion of the stability test was included in the manuscript as a new section (3.4 Stability of the catalysts).

4) It will be good to see XPS data of the optimal composites before and after HOR activity (after several cycles). 

Authors do not have access to XPS instrument. Even if the authors have access to the instrument, performing XPS analyses before HOR activity is suitable, but performing after HOR activity is not appropriate, since the quantity of catalyst for the modification of the electrode for HOR test is in lower than 1 mg and the quantity of mass for XPS analyses is in the order of 500 mg. 

5) Comment on the reproducibility, i.e. how many samples per trial were tested, HOR activity reproduction, etc.   

The reproducibility test was measured for the best catalysts, in this regard, the HOR activity test was recorded by triplicated, obtaining the catalytic activity parameters, through a statistical analysis, the relative standard deviation of the E1/2 and Jlim were estimated for both catalysts. The discussion of this results are included from line 458 to line 463.

Authors would like to thank to the reviewer, since authors consider that the comments from the reviewer will allow to enhance the discussion in the manuscript, increasing the evidence that support that PtCo catalysts are an ideal alternative for AEMFC/HP applications.

All the changes in the manuscript are highlighted in red, in order to make a difference with the first version of the manuscript.

Reviewer 3 Report

The article shows an interesting approach to preparation of ultra-low Pt loading catalysts with potential application as novel material for crucial fuel-cells elements. This paper focuses on the preparation and investigation of physical chemical parameters of PtCo catalysts deposited on different carbonaceous supports. The special emphasis was put on electrochemical catalytic activity towards hydrogen oxidation reaction. The detailed and comprehensive interpretation of the results has been made. It was concluded that MWCNT support is the most effective one due to the lowest half-wave potential (E1/2) and the highest value of limiting current density (Jlim) corresponding to low activation energy and large number of exposed active sites. Besides PtCo catalysts investigated the highest catalytic activity was evidenced for PtCo/MWCNT ones containing Pt in the range of 3-4 % i.e. over 5-folds lower comparing with conventional Pt/C catalyst. Authors also concluded that Pt content influences the HOR mechanism by modification of OH- interaction with surface Co species.

This work follows the scope of Nanomaterials and the manuscript supports reliable data and the results are discussed with clarity and coherence. The topic is of great interest at present, both from the scientific and technological point of view. The manuscript is well written but the points given below should be reconsidered and eventually corrected:

  • To replace “dec” with SI symbols: °C or K,
  • Line 164: “Since the main objective of this project …” – it is not clear what project do authors mean.
  • Line 190-191: “Thermogravimetric analysis (correct in plural “analyses”) were recorded in a TA instrument Q600 in order to determinate the metallic loading of the catalyst.”. Is TA really the method for reliable determination of materials’ chemical composition?
  • Line 196-197: “… the radio frequency was set up at 1300 W…” – is it correct to use W as frequency unit? Or power?
  • Line 439: “…MWCNT exhibited the lowest E1/2…” – my suggestion is to change to “…MWCNT-supported PtCo catalyst exhibited the lowest E1/2…”

For all the reasons above, I recommend acceptance of this manuscript and its publication in the Nanomaterials Journal after minor corrections.

Author Response

The article shows an interesting approach to preparation of ultra-low Pt loading catalysts with potential application as novel material for crucial fuel-cells elements. This paper focuses on the preparation and investigation of physical chemical parameters of PtCo catalysts deposited on different carbonaceous supports. The special emphasis was put on electrochemical catalytic activity towards hydrogen oxidation reaction. The detailed and comprehensive interpretation of the results has been made. It was concluded that MWCNT support is the most effective one due to the lowest half-wave potential (E1/2) and the highest value of limiting current density (Jlim) corresponding to low activation energy and large number of exposed active sites. Besides PtCo catalysts investigated the highest catalytic activity was evidenced for PtCo/MWCNT ones containing Pt in the range of 3-4 % i.e. over 5-folds lower comparing with conventional Pt/C catalyst. Authors also concluded that Pt content influences the HOR mechanism by modification of OH- interaction with surface Co species.

This work follows the scope of Nanomaterials and the manuscript supports reliable data and the results are discussed with clarity and coherence. The topic is of great interest at present, both from the scientific and technological point of view. The manuscript is well written but the points given below should be reconsidered and eventually corrected:

The reviewer performed a deep critical analysis of our manuscript, authors appreciate it the insight of the reviewer, in order to attend the observations from the reviewer point by point, authors response is highlighted in red:

  • To replace “dec” with SI symbols: °C or K,

The units of Tafel slope is expressed in mV dec-1, where dec implies the slope of a semi logarithmic plot of η vs Log j, is not related to a temperature scale unit.

  • Line 164: “Since the main objective of this project …” – it is not clear what project do authors mean.

The sentence was corrected accordingly to the reviewer, this is a research work, not a project.

  • Line 190-191: “Thermogravimetric analysis (correct in plural “analyses”) were recorded in a TA instrument Q600 in order to determinate the metallic loading of the catalyst.”. Is TA really the method for reliable determination of materials’ chemical composition?

The plural form was corrected in the manuscript. On the other hand, the composition of the bimetallic catalysts is determined by TG analyses and ICP-OES, first, the metallic loading is obtained by the thermal decomposition of the samples. Afterwards, the ashes of the TG analysis were diluted and analyzed by ICP-OES in order to determine the proportion of each metal. TG analysis is not suitable for the determination of the chemical composition of the catalyst by itself.

  • Line 196-197: “… the radio frequency was set up at 1300 W…” – is it correct to use W as frequency unit? Or power?

Done, the units of W are because this term is referred to the radio frequency power.

  • Line 439: “…MWCNT exhibited the lowest E1/2…” – my suggestion is to change to “…MWCNT-supported PtCo catalyst exhibited the lowest E1/2…”

Done, this correction was addressed, however, in this section, authors are discussing about the catalytic activity of the metallic catalyst in different supports, for that reason the correction was Pt/MWCNT.

For all the reasons above, I recommend acceptance of this manuscript and its publication in the Nanomaterials Journal after minor corrections.

Authors would like to thank to the reviewer, since his/her observation will allow to improve the quality of the manuscript.

Finally, the Authors would like to express their gratitude with the editor, because their close monitoring of our manuscript.

Round 2

Reviewer 2 Report

No more comments